# REFORMER: THE EFFICIENT TRANSFORMER

**Nikita Kitaev**[*]
U.C. Berkeley & Google Research
kitaev@cs.berkeley.edu

**Łukasz Kaiser**[*]
Google Research
{lukaszkaiser,levskaya}@google.com

**Anselm Levskaya**
Google Research

## ABSTRACT

Large Transformer models routinely achieve state-of-the-art results on a number of tasks but training these models can be prohibitively costly, especially on long sequences. We introduce two techniques to improve the efficiency of Transformers. For one, we replace dot-product attention by one that uses locality-sensitive hashing, changing its complexity from $O(L^2)$ to $O(L \log L)$, where $L$ is the length of the sequence. Furthermore, we use reversible residual layers instead of the standard residuals, which allows storing activations only once in the training process instead of $N$ times, where $N$ is the number of layers. The resulting model, the Reformer, performs on par with Transformer models while being much more memory-efficient and much faster on long sequences.

## 1 INTRODUCTION

The Transformer architecture (Vaswani et al., 2017) is widely used in natural language processing and yields state-of-the-art results on a number of tasks. To obtain these results, researchers have resorted to training ever larger Transformer models. The number of parameters exceeds 0.5B per layer in the largest configuration reported in (Shazeer et al., 2018) while the number of layers goes up to 64 in (Al-Rfou et al., 2018). Transformer models are also used on increasingly long sequences. Up to 11 thousand tokens of text in a single example were processed in (Liu et al., 2018) and when processing other modalities, like music (Huang et al., 2018) and images (Parmar et al., 2018), even longer sequences are commonplace. These large-scale long-sequence models yield great results but strain resources to the point where some argue that this trend is breaking NLP research[1]. Many large Transformer models can only realistically be trained in large industrial research laboratories and such models trained with model parallelism cannot even be fine-tuned on a single GPU as their memory requirements demand a multi-accelerator hardware setup even for a single training step.

Do large Transformer models fundamentally require such huge resources or are they simply inefficient? Consider the following calculation: the 0.5B parameters used in the largest reported Transformer layer account for 2GB of memory. Activations for 64K tokens with embedding size 1024 and batch size 8 account for $64K \times 1K \times 8 = 0.5B$ floats, requiring another 2GB of memory. If our memory use was only per-layer, then we should fairly easily fit a large Transformer even on sequences of length 64K on a single accelerator. Further, the whole corpus used to train BERT only requires 17GB to store. Why is it then that we cannot even fine-tune these models on single machines?

The above estimate includes only per-layer memory and input activations cost and does not take into account the following major sources of memory use in the Transformer.

- Memory in a model with $N$ layers is $N$-times larger than in a single-layer model due to the fact that activations need to be stored for back-propagation.
- Since the depth $d_{ff}$ of intermediate feed-forward layers is often much larger than the depth $d_{model}$ of attention activations, it accounts for a large fraction of memory use.
- Attention on sequences of length $L$ is $O(L^2)$ in both computational and memory complexity, so even for a single sequence of 64K tokens can exhaust accelerator memory.

---

[*]Equal Contribution
[1]https://hackingsemantics.xyz/2019/leaderboards/

We introduce the Reformer model which solves these problems using the following techniques:

- Reversible layers, first introduced in Gomez et al. (2017), enable storing only a single copy of activations in the whole model, so the $N$ factor disappears.

- Splitting activations inside feed-forward layers and processing them in chunks removes the $d_{ff}$ factor and saves memory inside feed-forward layers.

- Approximate attention computation based on locality-sensitive hashing replaces the O($L^2$) factor in attention layers with O($L \log L$) and so allows operating on long sequences.

We study these techniques and show that they have negligible impact on the training process compared to the standard Transformer. Splitting activations in fact only affects the implementation; it is numerically identical to the layers used in the Transformer. Applying reversible residuals instead of the standard ones does change the model but has a negligible effect on training in all configurations we experimented with. Finally, locality-sensitive hashing in attention is a more major change that can influence the training dynamics, depending on the number of concurrent hashes used. We study this parameter and find a value which is both efficient to use and yields results very close to full attention.

We experiment on a synthetic task, a text task (enwik8) with sequences of length 64K and an image generation task (imagenet-64 generation) with sequences of length 12K. In both cases we show that Reformer matches the results obtained with full Transformer but runs much faster, especially on the text task, and with orders of magnitude better memory efficiency.

## 2    LOCALITY-SENSITIVE HASHING ATTENTION

**Dot-product attention.** The standard attention used in the Transformer is the scaled dot-product attention (Vaswani et al., 2017). The input consists of queries and keys of dimension $d_k$, and values of dimension $d_v$. The dot products of the query with all keys are computed, scaled by $\sqrt{d_k}$, and a softmax function is applied to obtain the weights on the values. In practice, the attention function on a set of queries is computed simultaneously, packed together into a matrix $Q$. Assuming the keys and values are also packed together into matrices $K$ and $V$, the matrix of outputs is defined as:

$$\text{Attention}(Q, K, V) = \text{softmax}(\frac{QK^T}{\sqrt{d_k}})V \tag{1}$$

**Multi-head attention.** In the Transformer, instead of performing a single attention function with $d_{model}$-dimensional keys, values and queries, one linearly projects the queries, keys and values $h$ times with different, learned linear projections to $d_k$, $d_k$ and $d_v$ dimensions, respectively. Attention is applied to each of these projected versions of queries, keys and values in parallel, yielding $d_v$-dimensional output values. These are concatenated and once again projected, resulting in the final values. This mechanism is known as multi-head attention.

**Memory-efficient attention.** To calculate the memory use of the attention mechanism, let us focus on the attention computation from Equation 1. Let us assume that Q, K and V all have the shape $[batch\_size, length, d_{model}]$. The main issue is the term $QK^T$, which has the shape $[batch\_size, length, length]$. In the experimental section we train a model on sequences of length $64K$ – in this case, even at batch-size of 1, this is a $64K \times 64K$ matrix, which in 32-bit floats would take 16GB of memory. This is impractical and has hindered the use of the Transformer for long sequences. But it is important to note that the $QK^T$ matrix does not need to be fully materialized in memory. The attention can indeed be computed for each query $q_i$ separately, only calculating $\text{softmax}(\frac{q_i K^T}{\sqrt{d_k}})V$ once in memory, and then re-computing it on the backward pass when needed for gradients. This way of computing attention may be less efficient but it only uses memory proportional to $length$. We use this memory-efficient implementation of attention to run the full-attention baselines presented in the experimental section.

**Where do Q, K, V come from?** The multi-head attention described above operates on keys, queries and values, but usually we are only given a single tensor of activations A of the shape $[batch\_size, length, d_{model}]$ – e.g., coming from embedding the tokens in a sentence into vectors.

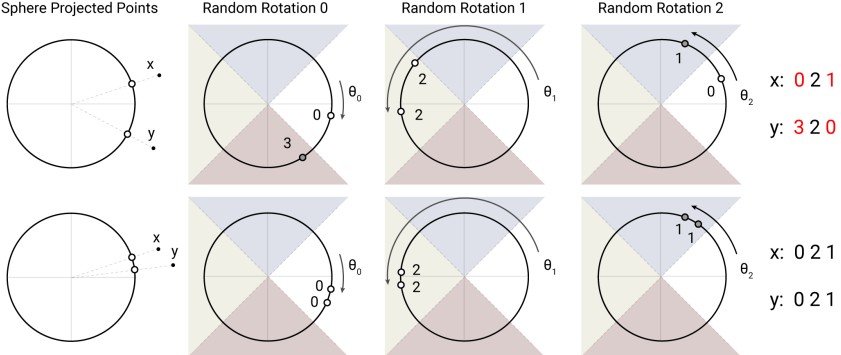

Figure 1: An angular locality sensitive hash uses random rotations of spherically projected points to establish buckets by an argmax over signed axes projections. In this highly simplified 2D depiction, two points $x$ and $y$ are unlikely to share the same hash buckets (above) for the three different angular hashes unless their spherical projections are close to one another (below).

To build Q, K and V from A, the Transformer uses 3 different linear layers projecting A into Q, K and V with different parameters. For models with LSH attention, we want queries and keys (Q and K) to be identical. This is easily achieved by using the same linear layer to go from A to Q and K, and a separate one for V. We call a model that behaves like this a shared-QK Transformer. It turns out that sharing QK does not affect the performance of Transformer, even if we additionally normalize the length of the keys K, as we show in the experimental Section 5.

**Hashing attention.** For the LSH attention, we start with two tensors, Q=K and V of the shape $[batch\_size, length, d_{model}]$. We keep the multi-head mechanism intact and focus on the attention computation from Equation 1. As already mentioned, the main issue is the term $QK^T$, which has the shape $[batch\_size, length, length]$. But note that we are actually only interested in $\text{softmax}(QK^T)$. Since softmax is dominated by the largest elements, for each query $q_i$ we only need to focus on the keys in K that are closest to $q_i$. For example, if K is of length 64K, for each $q_i$ we could only consider a small subset of, say, the 32 or 64 closest keys. That is much more efficient, but how can we find the nearest neighbors among the keys?

**Locality sensitive hashing.** The problem of finding nearest neighbors quickly in high-dimensional spaces can be solved by locality-sensitive hashing (LSH). A hashing scheme that assigns each vector $x$ to a hash $h(x)$ is called locality-sensitive if nearby vectors get the same hash with high probability and distant ones do not. In our case, we actually only require that nearby vectors get the same hash with high probability and that hash-buckets are of similar size with high probability.

We achieve this by employing random projections as follows (see Figure 1). To get $b$ hashes, we first fix a random matrix $R$ of size $[d_k, b/2]$. We then define $h(x) = \arg\max([xR; -xR])$ where $[u; v]$ denotes the concatenation of two vectors. This method is a known LSH scheme (Andoni et al., 2015) and is easy to implement and apply to batches of vectors.

**LSH attention.** Knowing our LSH scheme and the general idea of hashing attention, we will now formalize the LSH attention we use in this paper. We first rewrite the equation for normal attention, (1), for a single query position $i$ at a time:

$$o_i = \sum_{j \in \mathcal{P}_i} \exp\left(q_i \cdot k_j - z(i, \mathcal{P}_i)\right) v_j \qquad \text{where } \mathcal{P}_i = \{j : i \geq j\} \qquad (2)$$

We introduce the notation $\mathcal{P}_i$ to represent the set that the query at position $i$ attends to, and $z$ to denote the partition function (i.e. the normalizing term in the softmax). For clarity, we also omit scaling by $\sqrt{d_k}$.

For batching purposes we typically perform attention over a larger set $\widetilde{\mathcal{P}}_i = \{0, 1, \ldots, l\} \supseteq \mathcal{P}_i$ while masking out elements not in $\mathcal{P}_i$:

$$o_i = \sum_{j \in \widetilde{\mathcal{P}}_i} \exp\left(q_i \cdot k_j - m(j, \mathcal{P}_i) - z(i, \mathcal{P}_i)\right) v_j \quad \text{where } m(j, \mathcal{P}_i) = \begin{cases} \infty & \text{if } j \notin \mathcal{P}_i \\ 0 & \text{otherwise} \end{cases} \quad (3)$$

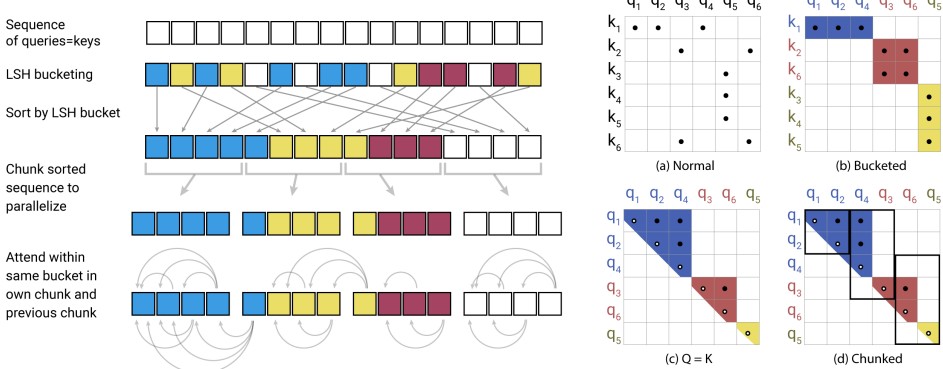

Figure 2: Simplified depiction of LSH Attention showing the hash-bucketing, sorting, and chunking steps and the resulting causal attentions. (a-d) Attention matrices for these varieties of attention.

Now we turn to LSH attention, which we can think of in terms of restricting the set $\mathcal{P}_i$ of target items a query position $i$ can attend to, by only allowing attention within a single hash bucket.

$$\mathcal{P}_i = \{j : h(q_i) = h(k_j)\} \tag{4}$$

Figure 2(a-b) shows a schematic comparison of full-attention with a hashed variant. Part (a) depicts that the attention matrix for full attention is typically sparse, but the computation does not take advantage of this sparsity. In (b), the queries and keys have been sorted according to their hash bucket. Since similar items fall in the same bucket with high probability, the full attention pattern can be approximated by only allowing attention within each bucket.

Hash buckets in this formulation tend to be uneven in size, which makes it difficult to batch across buckets. Moreover, the number of queries and the number of keys within a bucket may be unequal – in fact, it is possible for a bucket to contain many queries but no keys. To alleviate these issues, we first ensure that $h(k_j) = h(q_j)$ by setting $k_j = \frac{q_j}{\|q_j\|}$. Next, we sort the queries by bucket number and, within each bucket, by sequence position; this defines a permutation where $i \mapsto s_i$ after sorting. In the sorted attention matrix, pairs from the same bucket will cluster near the diagonal (as depicted in Figure 2c). We can follow a batching approach where chunks of $m$ consecutive queries (after sorting) attend to each other, and one chunk back (Figure 2d). Following our earlier notation, this corresponds to setting:

$$\widetilde{\mathcal{P}}_i = \left\{ j : \left\lfloor \frac{s_i}{m} \right\rfloor - 1 \leq \left\lfloor \frac{s_j}{m} \right\rfloor \leq \left\lfloor \frac{s_i}{m} \right\rfloor \right\} \tag{5}$$

If $\max_i |\mathcal{P}_i| < m$, then $\mathcal{P}_i \subseteq \widetilde{\mathcal{P}}_i$. In practice we set $m = \frac{2l}{n_{buckets}}$ (where $l$ is the sequence length). The average bucket size is $\frac{l}{n_{buckets}}$, and we assume that the probability of a bucket growing to twice that size is sufficiently low. The overall process of LSH attention is summarized in Figure 2.

**Multi-round LSH attention.** With hashing, there is always a small probability that similar items nevertheless fall in different buckets. This probability can be reduced by doing multiple rounds of hashing with $n_{rounds}$ distinct hash functions $\{h^{(1)}, h^{(2)}, \ldots\}$, such that:

$$\mathcal{P}_i = \bigcup_{r=1}^{n_{rounds}} \mathcal{P}_i^{(r)} \qquad \text{where } \mathcal{P}_i^{(r)} = \left\{ j : h^{(r)}(q_i) = h^{(r)}(q_j) \right\} \tag{6}$$

The multi-round case essentially involves performing LSH attention $n_{rounds}$ times in parallel; the details of the procedure are described in in Appendix A.

**Causal masking for shared-QK attention.** In a Transformer decoder, masking (denoted by $m(j, \mathcal{P}_i)$ in Equation 3) is used to prevent positions from attending into the future. To implement masking in LSH attention, we associate every query/key vector with a position index, re-order the position indices using the same permutations used to sort the query/key vectors, and then use a comparison operation to compute the mask.

Table 1: Memory and time complexity of attention variants. We write $l$ for length, $b$ for batch size, $n_h$ for the number of heads, $n_c$ for the number of LSH chunks, $n_r$ for the number of hash repetitions.

| Attention Type | Memory Complexity | Time Complexity |
|---|---|---|
| Scaled Dot-Product | $\max(bn_h ld_k, bn_h l^2)$ | $\max(bn_h ld_k, bn_h l^2)$ |
| Memory-Efficient | $\max(bn_h ld_k, bn_h l^2)$ | $\max(bn_h ld_k, bn_h l^2)$ |
| LSH Attention | $\max(bn_h ld_k, bn_h ln_r(4l/n_c)^2)$ | $\max(bn_h ld_k, bn_h n_r l(4l/n_c)^2)$ |

Table 2: Accuracies on the duplication task of a 1-layer Transformer model with full attention and with locality-sensitive hashing attention using different number of parallel hashes.

| Train \ Eval | Full Attention | LSH-8 | LSH-4 | LSH-2 | LSH-1 |
|---|---|---|---|---|---|
| Full Attention | 100% | 94.8% | 92.5% | 76.9% | 52.5% |
| LSH-4 | 0.8% | 100% | 99.9% | 99.4% | 91.9% |
| LSH-2 | 0.8% | 100% | 99.9% | 98.1% | 86.8% |
| LSH-1 | 0.8% | 99.9% | 99.6% | 94.8% | 77.9% |

While attention to the future is not allowed, typical implementations of the Transformer *do* allow a position to attend to *itself*. Such behavior is undesirable in a shared-QK formulation because the dot-product of a query vector with itself will almost always be greater than the dot product of a query vector with a vector at another position. We therefore modify the masking to forbid a token from attending to itself, except in situations where a token has no other valid attention targets (e.g. the first token in a sequence).

## 2.1 ANALYSIS ON A SYNTHETIC TASK

To verify the performance of LSH attention and study its behavior, we start with the following synthetic task: duplicate a sequence of symbols. In this task, each training and testing example has the form $0w0w$ where $w \in \{1, \ldots, N\}^*$ is a sequence of symbols ranging from 1 to $N$ (we use $N = 127$ in our experiments). An example with the word $w$ of length 3 is given below.

| **Example:** | 0 | 19 | 113 | 72 | 0 | 19 | 113 | 72 |
|---|---|---|---|---|---|---|---|---|

To study LSH attention, we train a language model on examples of the above form where each $w$ is of length 511 (so the whole input $0w0w$ is of length 1024). As this is a language modeling task, we always predict the next symbol given all the previous ones, but we mask the loss and accuracy to only consider positions in the second half of the input, i.e., those that can actually be predicted.

The above task can be solved perfectly (to accuracy 100% and loss 0) by a 1-layer Transformer model. Note though, that it requires non-local attention lookups, so it cannot be solved by any model relying on sparse attention with a limited span. To make it easy and fast to train but similar to models used in NLP, we use a 1-layer Transformer with $d_{model} = d_{ff} = 256$ and 4 heads. We train it for 150K steps in 4 different settings: with full attention, LSH attention with $n_{rounds} = 1$, $n_{rounds} = 2$ and $n_{rounds} = 4$.

From the results summarized in Table 2 we see that a model trained with full attention can be immediately used with LSH attention, but at some loss of accuracy. When trained from scratch with LSH attention, the model trained with 4 hashes achieves almost perfect accuracy as well. Interestingly, the accuracy becomes perfect when evaluated with 8 hashes. It goes down when evaluated with 2 or 1 hashes. Models trained with less hashes show worse results but even the model trained with just 1 hash performs almost perfectly when evaluated with 8 hashes.

## 3   REVERSIBLE TRANSFORMER

As the above section shows, the complexity of attention can be reduced from square in length to linear, provided an approximation is acceptable. But it is clear from Table 1 that each field starts with a $b \cdot n_h \cdot l$ term: the $b \cdot n_h \cdot l \cdot d_k$, or alternatively $b \cdot l \cdot d_{model}$ cost cannot be avoided. Indeed, the activations before each layer are already of the size $b \cdot l \cdot d_{model}$, so the memory use of the whole model with $n_l$ layers is at least $b \cdot l \cdot d_{model} \cdot n_l$. Even worse: inside the feed-forward layers of Transformer this goes up to $b \cdot l \cdot d_{ff} \cdot n_l$. In a big Transformer it is usual to set $d_{ff} = 4K$ and $n_l = 16$ so with $l = 64K$ this again would use an impractical $16GB$ of memory

In this section, we show how to reduce this cost by first dealing with the $n_l$ part of the term using reversible layers and then showing how chunking can allow us to handle the $d_{ff}$ problem. The effects of each of these approaches on memory and time complexity are summarized in Table 3.

**RevNets.** Reversible residual networks were introduced by Gomez et al. (2017) where it was shown that they can replace ResNets for image classification. The main idea is to allow the activations at any given layer to be recovered from the activations at the following layer, using only the model parameters. Rather than having to checkpoint intermediate values for use in the backward pass, layers can be reversed one-by-one as back-propagation proceeds from the output of the network to its input. Whereas a normal residual layer performs a function $x \mapsto y$ that operates on a single input and produces a single output and has the form $y = x + F(x)$, a reversible layer works on pairs of inputs/outputs: $(x_1, x_2) \mapsto (y_1, y_2)$, and follows the equations:

$$y_1 = x_1 + F(x_2) \qquad\qquad y_2 = x_2 + G(y_1) \qquad (7)$$

A layer can be reversed by subtracting (rather than adding) the residuals:

$$x_2 = y_2 - G(y_1) \qquad\qquad x_1 = y_1 - F(x_2) \qquad (8)$$

**Reversible Transformer.** We apply the RevNet idea to the Transformer by combining the attention and feed-forward layers inside the revnet block. In the notation above, F becomes an attention layer while G becomes the feed-forward layer. Note that Layer Normalization (Ba et al., 2016) is moved inside the residual blocks.

$$Y_1 = X_1 + \text{Attention}(X_2) \qquad\qquad Y_2 = X_2 + \text{FeedForward}(Y_1) \qquad (9)$$

The reversible Transformer does not need to store activations in each layer and so gets rid of the $n_l$ term. In Section 5 we show that it performs the same as the normal Transformer when using the same number of parameters; we achieve this by having both $x_1$ and $x_2$ have size $d_{model}$.

**Chunking.** While reversibility covers the $n_l$ term, the thicker layers can still use a lot of memory. The feed-forward layer in particular can use intermediate vectors of dimensionality $d_{ff} = 4K$ or higher. However, computations in feed-forward layers are completely independent across positions in a sequence, so the computation can be split into $c$ chunks:

$$Y_2 = \left[ Y_2^{(1)}; \ldots; Y_2^{(c)} \right] = \left[ X_2^{(1)} + \text{FeedForward}(Y_1^{(1)}); \ldots; X_2^{(c)} + \text{FeedForward}(Y_1^{(c)}) \right] \quad (10)$$

This layer is typically batched by performing operations for all positions in parallel, but operating on one chunk at a time can reduce memory. The reverse computation in (8) and the backward pass are also chunked. In addition to the feed-forward layers, for models with large vocabulary (more than $d_{model}$ word types) we also chunk the log-probabilities at the output and calculate the loss for sections of the sequence at a time.

**Chunking, large batches and parameter reuse.** With chunking and reversible layers the memory we use for activations in the whole network is independent of the number of layers. The same is not true for parameters though as their number grows with the number of layers. This problem is remedied though because we can swap layer parameters to and from CPU memory when this layer is not computing. In a standard Transformer this would be inefficient because memory transfer to CPU is slow. The batch size multiplied by length in Reformer is much larger though and therefore the amount of compute done with the parameters amortizes the cost of their transfer.

Table 3: Memory and time complexity of Transformer variants. We write $d_{model}$ and $d_{ff}$ for model depth and assume $d_{ff} \geq d_{model}$; $b$ stands for batch size, $l$ for length, $n_l$ for the number of layers. We assume $n_c = l/32$ so $4l/n_c = 128$ and we write $c = 128^2$.

| Model Type | Memory Complexity | Time Complexity |
|---|---|---|
| Transformer | $\max(bld_{ff}, bn_h l^2)n_l$ | $(bld_{ff} + bn_h l^2)n_l$ |
| Reversible Transformer | $\max(bld_{ff}, bn_h l^2)$ | $(bn_h ld_{ff} + bn_h l^2)n_l$ |
| Chunked Reversible Transformer | $\max(bld_{model}, bn_h l^2)$ | $(bn_h ld_{ff} + bn_h l^2)n_l$ |
| LSH Transformer | $\max(bld_{ff}, bn_h ln_r c)n_l$ | $(bld_{ff} + bn_h n_r lc)n_l$ |
| Reformer | $\max(bld_{model}, bn_h ln_r c)$ | $(bld_{ff} + bn_h n_r lc)n_l$ |

## 4 RELATED WORK

The Transformer model introduced in (Vaswani et al., 2017) has been used widely in natural language tasks and further extended to model diverse data such as music scores (Huang et al., 2018), and images (Parmar et al., 2018; Ramachandran et al., 2019). Most notably, this model class has been applied successfully in the self-supervised training of extremely large language models (Devlin et al., 2018; Radford et al., 2019).

Given the enormous computational requirements of state of the art sequence models, there has been increasing interest in finding methods to reduce the memory footprint and computational requirements of Transformer models. In addition to standard methods such as precision reduction and gradient checkpointing (Sohoni et al., 2019), more efficient versions of the Transformer model's self-attention mechanism (Sukhbaatar et al., 2019a;b) have also recently been explored.

In particular, leveraging sparsity in the attention layers has proved fruitful. OpenAI introduced the sparse Transformer (Child et al., 2019) which exploits a factorized sparse representation of attention. Using product-key attention to increase the key space has also been used to reduce memory requirements in the feed-forward layers with no loss in performance (Lample et al., 2019).

Locality-sensitive hashing (LSH) has, to our knowledge, not been directly applied to Transformer attention layers before. But previous work using external memory with neural networks has dealt with memories of large sizes. The original implementation of memory networks (Weston et al., 2014) and later work on scaling it (Bordes et al., 2015; Chandar et al., 2016) used memory with size in the millions. The cost of doing so is that the memory must be fixed prior to training. Moreover, since during the beginning of training the model is unlikely to query the memory correctly, strong supervision is used to encourage the model to query memory locations that are useful. These hints are either given as additional supervising information by the task or determined heuristically as in Hill et al. (2015). The requirement that the memory be fixed before has been removed in Santoro et al. (2016) at the cost of memory size and later alleviated by Rae et al. (2016). The last paper considered memory lookups with approximate nearest neighbors including both LSH and random kd-trees, but only for lookups in external memory.

## 5 EXPERIMENTS

In this section we present experimental results demonstrating the techniques described above. We analyze the techniques one-by-one to make clear which combinations have impact on performance. We start by showing that reversible layers and shared query-key spaces do not impact performance, then proceed to analyze hashing attention and finally the full Reformer model.

We ran our experiments on the imagenet64 and enwik8-64K tasks, where the latter is a variant of enwik8 that is chunked into subsequences of $2^{16} = 64K$ tokens. We use 3-layer models for our ablations so as to make it tractable to compare with the regular Transformer, which has high memory usage and performs full $O(l^2)$ attention. All experiments have $d_{model} = 1024$, $d_{ff} = 4096$, $n_{heads} = 8$, and a total batch size of 8 sequences. We used the Adafactor optimizer (Shazeer & Stern, 2018) for training these models. We also evaluate on the WMT 2014 English-to-German translation task, following the hyperparameters of Vaswani et al. (2017). Training for all experiments

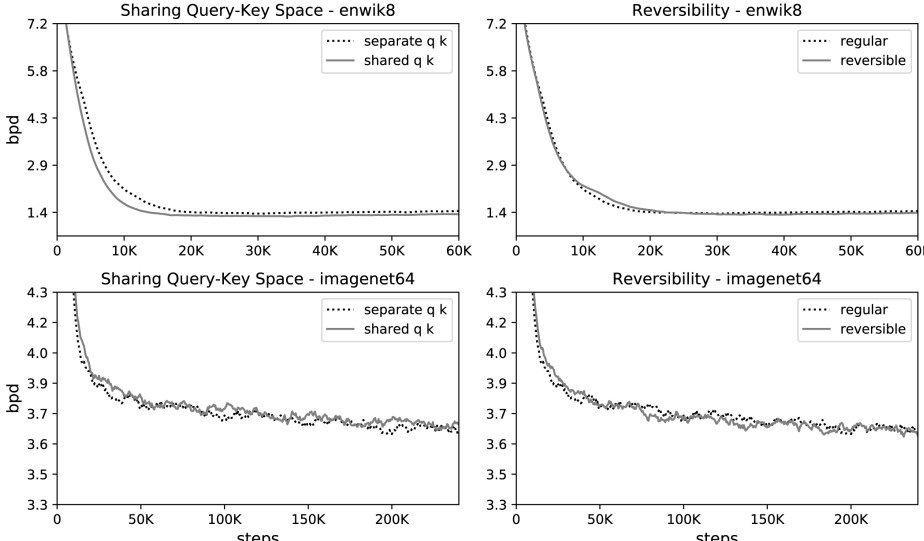

Figure 3: Effect of shared query-key space (left) and reversibility (right) on performance on enwik8 and imagenet64 training. The curves show bits per dim on held-out data.

Table 4: BLEU scores on newstest2014 for WMT English-German (En–De). We additionally report detokenized BLEU scores as computed by sacreBLEU (Post, 2018).

| Model | BLEU | *sacreBLEU* | |
| | | *Uncased*[3] | *Cased*[4] |
|---|---|---|---|
| Vaswani et al. (2017), base model | 27.3 | | |
| Vaswani et al. (2017), big | 28.4 | | |
| Ott et al. (2018), big | 29.3 | | |
| Reversible Transformer (base, 100K steps) | 27.6 | *27.4* | *26.9* |
| Reversible Transformer (base, 500K steps, no weight sharing) | 28.0 | *27.9* | *27.4* |
| Reversible Transformer (big, 300K steps, no weight sharing) | 29.1 | *28.9* | *28.4* |

was parallelized across 8 devices (8 GPUs or 8 TPU v3 cores). Code for training our models is made publicly available.[2]

**Effect of sharing QK.** We first consider the effect of shared-QK attention on a regular Transformer model. Shared-QK attention sets $k_j = \frac{q_j}{\|q_j\|}$ and prevents tokens from attending to themselves (except when no other context is available). In the left part of Figure 3, we plot perplexity curves for both regular and shared-QK attention. A shared query-key space does not perform worse than regular attention; in fact, for enwik8 it appears to train slightly faster. In other words, we are not sacrificing accuracy by switching to shared-QK attention.

**Effect of reversible layers.** In the two plots on the right in Figure 3, we compare a regular Transformer per Vaswani et al. (2017) with the reversible one describe in Section 3. The two models have identical parameter counts, and the learning curves likewise appear to be nearly the same. These results show that the memory savings in the reversible Transformer do not come at the expense of accuracy.

**Reversible layers in machine translation.** We also evaluate reversible layers in the context of an encoder-decoder Transformer model for machine translation from English to German. We start by making both the encoder and the decoder fully reversible in the Transformer-base architecture, and

---

[2] https://github.com/google/trax/tree/master/trax/models/reformer

[3] BLEU+case.lc+lang.en-de+numrefs.1+smooth.exp+test.wmt14/full+tok.intl+version.1.4.3

[4] BLEU+case.mixed+lang.en-de+numrefs.1+smooth.exp+test.wmt14/full+tok.intl+version.1.4.3

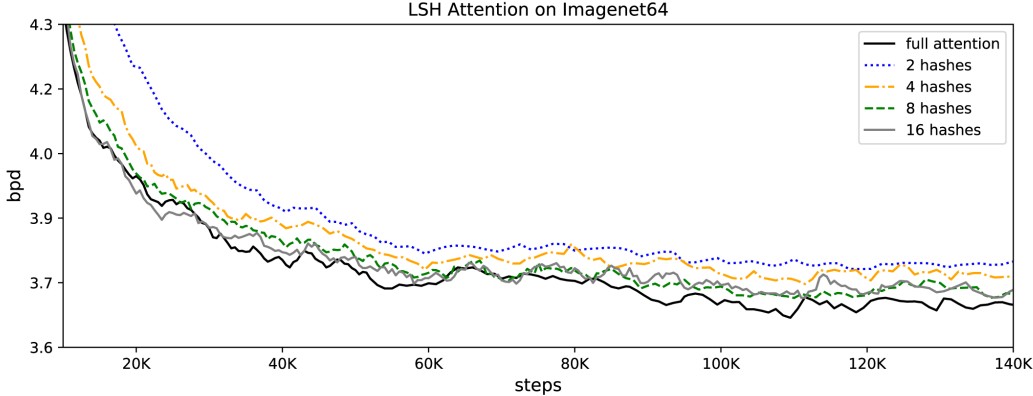

Figure 4: LSH attention performance as a function of hashing rounds on imagenet64.

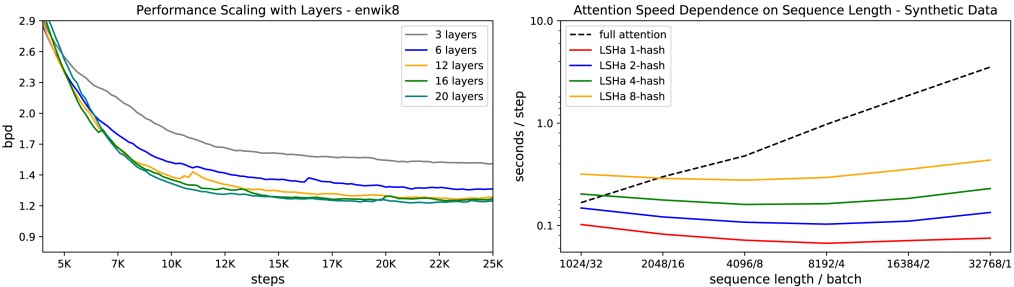

Figure 5: Left: LSH attention performance as a function of number of layers on enwik8. Right: Speed of attention evaluation as a function of input length for full- and LSH- attention.

see that the resulting model performs comparably to Vaswani et al. (2017) when trained for 100K steps. We also evaluate training for a greater number of steps and with a larger model. Reformer models are very memory-efficient, so for the latter two experiments we do not need to save memory by sharing embedding and output projection weight matrices throughout the model. Results are shown in Table 4. We do not apply LSH attention in this setting because examples are single sentences, and sentences tend to be relatively short. Our typical LSH attention configuration uses chunks of 128 tokens after hashing and sorting, whereas the examples in the WMT14 test set are all shorter than 128 tokens.

**LSH attention in Transformer.** LSH attention is an approximation for full attention that, as evidenced in Figure 4, becomes more accurate as the number of hashes increases. At $n_{rounds} = 8$, it already almost matches full attention. The computational cost of a model grows with the number of hashes, so this hyperparameter can be adjusted depending on the available compute budget. Additionally, as in Table 2, the number of hashes can be increased at evaluation time to produce more accurate results. On the right half of Figure 5, we plot the speed of different attention types vs. the sequence length, while holding the total number of tokens fixed. We see that while regular attention becomes slower at longer sequence length, LSH attention speed remains flat.

**Large Reformer models.** To verify that the Reformer can indeed fit large models on a single core and train fast on long sequences, we train up to 20-layer big Reformers on enwik8 and imagenet64. As can be seen in Figure 5, these models fit into memory and train. We were not able to train Transformer baselines in this case as they are too slow and memory-hungry, but we see clear improvement with the number of layers. A 12-layer model on enwik8 trained for 20K steps with a dropout rate of 0.1 achieves 1.19 bits/dim on the test set. We also trained a 12-layer Reformer model for longer with further tuning and improvements and we reached 1.05 bits/dim on the enwik8 test set.

## 6 CONCLUSION

Reformer combines the modeling capacity of a Transformer with an architecture that can be executed efficiently on long sequences and with small memory use even for models with a large number of layers. We believe that this will help large, richly-parameterized Transformer models become more widespread and accessible. Also, the ability to handle long sequences opens the way for the use of the Reformer on many generative tasks. In addition to generating very long coherent text, the Reformer can bring the power of Transformer models to other domains like time-series forecasting, music, image and video generation.

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

# A    MULTI-ROUND LSH ATTENTION

In this section we describe in more detail the multi-hash version of our LSH attention mechanism. We first repeat Equation (3) from the main text, which describes a general formulation of attention with sparsity:

$$o_i = \sum_{j \in \widetilde{\mathcal{P}}_i} \exp\left(q_i \cdot k_j - m(j, \mathcal{P}_i) - z(i, \mathcal{P}_i)\right) v_j \quad \text{where} \;\; m(j, \mathcal{P}_i) = \begin{cases} \infty & \text{if } j \notin \mathcal{P}_i \\ 0 & \text{otherwise} \end{cases} \quad (3)$$

In the multi-round case, a query position $i$ can attend to key positions $\mathcal{P}_i$ as defined in (6), which we also repeat here:

$$\mathcal{P}_i = \bigcup_{r=1}^{n_{rounds}} \mathcal{P}_i^{(r)} \qquad \text{where} \;\; \mathcal{P}_i^{(r)} = \left\{ j : h^{(r)}(q_i) = h^{(r)}(q_j) \right\} \qquad (6)$$

For batching purposes, attention is performed on chunks of sorted queries/keys:

$$\widetilde{\mathcal{P}}_i^{(r)} = \left\{ j : \left\lfloor \frac{s_i^{(r)}}{m} \right\rfloor - 1 \leq \left\lfloor \frac{s_j^{(r)}}{m} \right\rfloor \leq \left\lfloor \frac{s_i^{(r)}}{m} \right\rfloor \right\} \qquad (11)$$

Combining (3) and (6) gives:

$$o_i = \sum_{j \in \widetilde{\mathcal{P}}_i} \exp\left(q_i \cdot k_j - m(j, \mathcal{P}_i) - z(i, \mathcal{P}_i)\right) v_j \qquad (12)$$

$$= \sum_{r=1}^{n_{rounds}} \exp\left(z(i, \mathcal{P}_i^{(r)}) - z(i, \mathcal{P}_i)\right) \sum_{j \in \widetilde{\mathcal{P}}_i^{(r)}} \frac{1}{N_{i,j}} \exp\left(q_i \cdot k_j - m(j, \mathcal{P}_i^{(r)}) - z(i, \mathcal{P}_i^{(r)})\right) v_j$$
$$(13)$$

$$= \sum_{r=1}^{n_{rounds}} \exp\left(z(i, \mathcal{P}_i^{(r)}) - z(i, \mathcal{P}_i)\right) o_i^{(r)} \qquad (14)$$

$$o_i^{(r)} = \sum_{j \in \widetilde{\mathcal{P}}_i^{(r)}} \exp\left(q_i \cdot k_j - m_{i,j}^{(r)} - z(i, \mathcal{P}_i^{(r)})\right) v_j \qquad (15)$$

$$\text{where } N_{i,j} = \left| \left\{ r' : j \in \mathcal{P}_i^{(r')} \right\} \right| \text{ and } m_{i,j}^{(r)} = \begin{cases} \infty & \text{if } j \notin \mathcal{P}_i^{(r)} \\ 10^5 & \text{if } i = j \\ \log N_{i,j} & \text{otherwise} \end{cases} \qquad (16)$$

Each round of LSH attention produces a vector $o_i^{(r)}$ that can be computed independently from other rounds, except for the inclusion of a term $N_{i,j}$ to avoid double-counting elements when constructing the union of $\mathcal{P}_i^{(r)}$ sets. In our implementation we fold the $N_{i,j}$ factor into the masking term $m_{i,j}^{(r)}$.

We also modify $m_{i,j}^{(r)}$ to introduce a special case for $i = j$. This case is added because causal masking in a standard Transformer allows position $i$ to attend to itself, which is not desirable in a shared-QK formulation. We set the mask to a large but finite value to disallow attention-in-place, except in the situation where a token has no other valid attention targets. For example, the first token in a sequence attends only to itself, because no prior context is available.

