# OpenReview forum: "Reformer: The Efficient Transformer"
_ICLR.cc/2020/Conference — Accept (Talk)_

### Official Review · AnonReviewer3 · 2019-10-22
**Official Blind Review #3**

**Rating:** 6

**Review:**

This paper presents an attempt to reduce the memory complexity of Transformers. The authors call their model the Reformer. It presents a LSH based self-attention mechanism, along with reversible adaptation of Transformers. The Locality sensitive hashing scheme reduces complexity from L^2 to L which is pretty neat.

Tackling the quadratic complexity of self-attention is indeed an important and nice direction. I think the LSH based attention quite novel and is a natural solution to reducing the complexity of the self-attention module.  However, I think the technical description could be improved as the current form is quite confusing and difficult to parse.

The experiments are a little on the weaker side. Authors presented results on imagenet, enwiki and a synthetic task. I am mainly concerned if the Reformer works on tasks such as machine translation or other NLP tasks. The paper does not present much evidence that the effectiveness of LSH is broad and versatile.

My current vote is a weak accept, based on some preliminary understanding and the general novelty of the idea.

I do have some questions/issues/comments:

1) Given that there is some form of QK sorting, how is it possible to mask the future? Is this because tokens are sorted within buckets?
2) Can the authors clarify what "Causal masking on the Transformer is typically implemented to allow a position i to attend to itself." mean?
3) I'm a little confused about how the sorting is being done. Can this be done in an end-to-end differentiable manner?
4) Can the authors present some results on other tasks? While neat, I think other tasks (e.g., MT or QA) can be investigated to further ascertain that the LSH attention works well. Current experimental results are not too convincing.


**Experience Assessment:**

I have published one or two papers in this area.

**Review Assessment: Checking Correctness Of Derivations And Theory:**

N/A

**Review Assessment: Checking Correctness Of Experiments:**

I assessed the sensibility of the experiments.

**Review Assessment: Thoroughness In Paper Reading:**

I read the paper at least twice and used my best judgement in assessing the paper.

---

> ### Author Response · Authors · 2019-11-14
> **Thank you for your comments and questions**
>
> Thank you for your feedback and questions regarding the paper, which we address one-by-one below. We’re updated the technical sections of the paper to increase clarity; please let us know if there are still any sections that you find difficult to parse.
>
> 1. How is causal masking implemented?
>
> To mask out attention to the future, we associate each query/key vector with a position index, where the position indices are then sorted using the same permutation as the QK sort. Position indices are compared for each query-key dot product, and the attention probability is masked to zero if the query comes before the key.
>
> 2. Attention-in-place
>
> Thank you for pointing out that this was unclear. We have updated the paper to elaborate on this point.
>
> In a typical Transformer implementation, positions can attend to themselves. There is a dot product between the query vector at position i and the key vector at position i; if this dot product is high then the value vector at position i will contribute to the output of the attention layer. This behavior isn’t very useful because local information is already propagated through the residual connections, but standard attention can learn to drive this attention probability to zero by making q_i and k_i orthogonal. Shared-QK attention, on the other hand, can’t reduce this weight because the query and the key are the same vector. To address this issue, we don’t allow attention-in-place for the Reformer.
>
> 3. Backprop through LSH attention, and sorting.
>
> We use sorting as an implementation for allowing items that map to the same hash bucket to attend to each other. Similar items get mapped to the same hash bucket with high probability, which allows similar item pairs to participate in both the forward and backward passes. Each hash bucket may contain a certain number of unrelated items, in which case there will be a gradient signal that either up-weighs or down-weighs attention to these items.
>
> We don’t differentiate through the hash bucket assignment procedure, or the choice of what order to sort the items into. Rather, these operations take query/key vectors as input where LSH maps nearby vectors to the same bucket with high probability. Therefore, the sorting re-adjusts any time parameter updates to cause relevant vector pairs to have higher dot product, and “unhelpful” vector pairs to have lower dot products.
>
> 4. Additional tasks.
>
> Thank you for your recommendation that we evaluate on other tasks. Prompted by your recommendation we started working on applying the Reformer to machine translation (we didn’t do that before since sequences are short in translation data-sets so it was not a prime target for Reformer).  Thus far we have trained a decoder-only Reformer on concatenated English-then-German sentence pairs, and we do not observe any difference compared to a regular Transformer LM. We’re in the process of constructing and tuning a more standard encoder-decoder approach that likewise uses the Reformer architecture. In the final version of our paper, we’ll report BLEU numbers and comparisons for English-German translation -- the current runs make us believe that they will be the same as for Transformer.

---

### Official Review · AnonReviewer1 · 2019-10-24
**Official Blind Review #1**

**Rating:** 8

**Review:**

This paper presents a method to make Transformer models more efficient in time and memory. The proposed approach consists mainly of three main operations:
- Using reversible layers (inspired from RevNets) in order to prevent the need of storing the activations of all layers to be reused for back propagation;
- Using locality sensitive hashing to approximate the costly softmax(QK^T) computation in the full dot-product attention;
- Chunking the feed-forward layers computations to reduce their cost.
This approach is first applied to a toy dataset to analyze its complexity, then tested on enwik8 language modelling task and imagenet-64 image generation task for ablation study and performance assessment.

The problem approached by the paper is interesting and the proposed approach is novel to the best of my knowledge. The paper is well structured and clearly written a part from some small typos (see minor comments below).

While the analysis of complexity is sound and convincing, and the fact of being able to train larger Reformers is very interesting, I have some questions and concerns about the approach and experiments.
- Effect of reversible layers: It is clear for the experiment of Imagenet64 that the effect is negligible, but the experiment on enwik8 in the paper seems unfinished. Did the authors manage to finish the training, and does it confirm the observation?
- Sharing QK: I am a bit confused about the effect and usefulness of this operation. Can the authors comment on why it is needed for LSH attention? It seems to me that the same operations can be achieved with different Q and K. Indeed, doing so, the authors slightly reduce the capacity of the model. The observed non-significantly decreased performance can be an effect of using only 3-layers. This may explain why the results reported for larger models in figure 5 show higher bpc than similar size state of the art models.
- Time per iterations: Can the authors report the time per iteration for the larger hash rounds (8 and 16) that are closer to full attention? For the highest reported number (4), from a quick and not precise look at figure 4, it seems that the performance achieved by the proposed method after 140k iterations is achieved by the full attention after ~40k iterations. The gain in time per iteration for this particular number of hash rounds can be lost by the loss in performance.
- Can the authors detail how they chose the hyperparameters of their approach? e.g. the size of hash buckets, the distribution used to generate the random matrix R ..
- The reported results can be made stronger by reporting average/error bars across several trial to show consistency.

Minor: typos:
Dimension of matrix R [d_k, d_b/2] -> [d_k, b/2]
Last paragraph of page 6: state of these art -> state of the art

———————————————
After rebuttal:
I have read the authors answer, and found they addressed my concerns. I'm therefore increasing my score.

**Experience Assessment:**

I have read many papers in this area.

**Review Assessment: Checking Correctness Of Derivations And Theory:**

I assessed the sensibility of the derivations and theory.

**Review Assessment: Checking Correctness Of Experiments:**

I carefully checked the experiments.

**Review Assessment: Thoroughness In Paper Reading:**

I read the paper at least twice and used my best judgement in assessing the paper.

---

> ### Author Response · Authors · 2019-11-14
> **Thank you for your thoughtful feedback**
>
> We thank the reviewer for thoughtful feedback on our paper. We have posted an update to address some of the comments, which we detail below.
>
> 1. Effect of reversible layers
>
> We updated the figures in the paper to cover longer training durations. As expected, reversible layers perform the same as regular Transformer layers on enwik8.
>
> 2. Sharing QK
>
> This operation is needed so that we can batch LSH attention on current hardware. Absent any hardware requirements, we could do unshared LSH attention as illustrated in Figure 2(b). Each hash bucket in the unshared condition may contain a different number of queries, a different number of keys, and moreover there is no relationship between the number of queries and the number of keys. Computing one bucket at a time would be too slow, and it’s unclear how to batch buckets of highly variable sizes. With shared-QK, as in Figure 2(c-d), we can batch effectively because the entries we want to calculate cluster near the main diagonal (after sorting). Let us stress though that this is purely a speed optimization which we did due to the realities of current hardware architectures. It works, but one could indeed hope that one day it will not be necessary.
>
> 3. Enwik8 results
>
> We’re happy to report that, with further tuning, our 12-layer model reaches 1.05 bits/dim on enwik8. Adjusting optimizer settings and dropout played a big role in improving perplexity for this task.
>
> 3. Time per iterations
>
> Thank you for your suggestion. We’ve updated the right part of Figure 5 to sweep over a larger range of hash numbers and sequence lengths. Although full attention is fast for short sequences, its O(n^2) scaling makes it rather slow at long sequence lengths, even when compared to the 8-hash LSH variant.
>
> 4. Hyperparameters
>
> The random matrix R has i.i.d. unit Gaussian entries, following Andoni et al. (https://arxiv.org/pdf/1509.02897.pdf; page 4). The number of hash buckets was chosen such that each bucket would have 64 entries on average. Making the hash buckets smaller hurts accuracy, whereas increasing it doesn’t seem to do much other than making the model slower.
>
> 5. Variance between runs.
>
> Thank you for your pointing this out. For now, we can report that the variance between runs, at convergence, is minimal: we see no variance when rounding to two decimal points.

---

### Official Review · AnonReviewer2 · 2019-11-04
**Official Blind Review #2**

**Rating:** 8

**Review:**

This manuscript presents a number of algorithmic techniques to reduce the computational and space complexity of Transformer, a powerful and very popular deep learning model for natural language processing (NLP). Although Transformer has revolutionized the field of NLP, many small groups cannot make a full use of it due to lack of necessary computational resources. As such, it is very important to improve the space and computational complexity of this popular deep model. The techniques presented in this manuscript seem to be very reasonable and the experimental results also indicate that they are effective. My major concern is that the authors shall present more detailed experimental results. In addition to bits per dim, it will also better if the authors can evaluate the performance in terms of other metrics.

**Experience Assessment:**

I have read many papers in this area.

**Review Assessment: Checking Correctness Of Derivations And Theory:**

I did not assess the derivations or theory.

**Review Assessment: Checking Correctness Of Experiments:**

I assessed the sensibility of the experiments.

**Review Assessment: Thoroughness In Paper Reading:**

I read the paper at least twice and used my best judgement in assessing the paper.

---

> ### Author Response · Authors · 2019-11-14
> **Thank you for your feedback**
>
> We thank the reviewer for feedback and comments on our paper. We have updated the paper to address some concerns and we’re working on preparing additional experiments and results to more thoroughly characterize the behavior of the proposed method, which will address all other questions.
>
> We posted a revised version of the paper with updated results figures. In particular, we’ve completed the curves and updated our illustration of the wall clock time used by different attention methods. This makes it clearer at what length the LSH attention starts saving time compared to full attention and at which number of hashes (Figure 5).
>
> As for the question on metrics: we will expand the results to include machine translation in the final version (we didn’t do this initially since sequences are quite short in translation datasets and as such don’t make for ideal targets for the Reformer). We did not get the complete results yet, but we started training a Reformer language model on concatenated English-then-German sentence pairs and we do not observe any major difference compared to a regular Transformer LM. We are also putting together and tuning a more conventional encoder-decoder approach that uses the Reformer architecture and we will include a comparison of BLEU between such Reformer and the Transformer in the final version of our paper.
>
> We are also happy to report that, with further tuning, a 12-layer Reformer model can achieve 1.05 bits/dim on the enwik8 test set. In terms of other metrics, this corresponds to 77.8% byte-level accuracy.

---

### Public Comment · ~Jack_William_Rae1 · 2019-10-01
**Query about shared queries**

Hello, I enjoyed reading your paper and think this area of research is very exciting. One minor concern/query that I had, which hopefully can be addressed by the time of author response / paper update, was why the results on enwik8 are so far from prior published transformer results. E.g. there have been transformers published with 0.99bpc (TransformerXL), and several others around the 1.0-1.06 mark. From Figure 5 it appears as though the models will not obtain results lower than 1.2 bpc. Is it because the models have not converged, or is it because the test data is different (i.e. you don't use the 90MB/5MB/5MB train/valid/test split)?

Furthermore I was not able to replicate the positive benefit of sharing the key and query weight matrices. Namely, I used a 24 layer TransformerXL baseline --- exactly the same setup as the published paper --- which obtains 0.992 bpc and then tried tying the weights between the query and key parameters; this lead to a model with 1.012; which is 0.02bpc drop. The non-shared variant had a faster drop in training and validation learning curves. I don't mean for this to detract from the paper - but just to add another data-point on this observation. Perhaps you don't have to share the weights for queries and keys (still normalizing them of course, so you can use the spherical LSH).

---

> ### Author Response · Authors · 2019-10-01
> **Thank you for the information**
>
> Hi Jack, thank you very much for additional information! As for sharing queries and keys: we did see slower training at first just copying the hyperparameters, but it reached the same accuracy later in training with appropriate learning rate. As for comparisons to SOTA results: we used 12 layers and a default Transformer configuration using the Adafactor optimizer without any tuning other than learning rate. Al Rfou et al. report 1.11bpc for a similar 12-layer configuration but with tuning, extra losses and a different optimizer, while the numbers you cite are for a highly-tuned model with 24 layers and a different architecture if I understand correctly. The purpose of our paper is to introduce new techniques and show they match the baseline Transformer perplexity with lower memory and training time, we leave extensions to other Transformer variants for future work (as there are quite many of them and more by the day).

---

### Public Comment · ~Aurko_Roy1 · 2019-10-02
**Complexity of LSH attention**

I am a bit confused by the complexity of LSH attention in Table 2. In particular, if the number of buckets is denoted by n_c then the total cost would be O(l^2/n_c) (average bucket occupancy) together with O(l*n_c) for the cost of computing the random projections. Do you include the latter in the total cost - i.e. the cost to compute the hashes?

---

> ### Author Response · Authors · 2019-10-02
> **Cost clarifications**
>
> Thanks you very much for your interest!
>
> If n_c stands for the number of hash buckets, then (as we explain in the paper) we will split the sequence into chunks of length l_c = 2l/n_c (since we use chunks twice the size of expected bucket). In most experiments we picked n_c so that l_c = 64. Note that we attend to the current and previous chunk, so with l_c = 64 we perform full 64x128 attentions, and there are l/64 of them. So the cost of that part is (2l/n_c)^2 but since we pick n_c so that l_c is constant, it can also be denoted simply by O(l), where the main constant factor is the 64x128 matrix multiplication and, more importantly, memory access.
>
> The above calculation, as you note, does indeed *not* include the computation of the hash id. Hash id is computed by multiplying activations of length l by a random matrix and picking the argmax. This is again of the order O(l) with the constant n_c, as you say. In theory, if n_c were very large, this could grow prohibitively. In that case one could use projection hashes -- e.g., multiply by 2 different matrices into size sqrt(n_c) and use the 2 hashes as higher and lower bits of the hash. In practice though, this matrix multiplication is quite cheap -- even upto n_c=1024 the cost of this matmul is negligible compared to the cost of memory access during hashing -- that's why we did not emphasize it in the analysis.

---

> > ### Public Comment · ~Aurko_Roy1 · 2019-10-02
> > **Thanks for the clarification**
> >
> > I see, thanks for the clarification! An alternative analysis could be O(l*n_c) (for computing hash via random projection) and O(l*l_c)=O(l^2/n_c) (for attention in the chunks), with total cost O(l*n_c + l^2/n_c). This expression could be minimized by choosing n_c = sqrt(l), and you would get total complexity O(l^{1.5}) as in Child et al [1]
> >
> > Cool idea though!
> >
> > [1] https://arxiv.org/abs/1904.10509

---

> > > ### Author Response · Authors · 2019-10-02
> > > **Update on asymptotic complexity vs implementation**
> > >
> > > Thank you very much for your calculation!
> > >
> > > The term you mentioned, O(l*n_c + l^2/n_c), is correct for the basic implementation we chose for practical reasons, to minimize constants rather than the asymptotic time. One of the most common hashes used throughout LSH though is based on plane cutting: after a product with N vectors the hash is N bits, the sign of these products. That yields n_c = 2^N hash buckets in time O(N) = O(log(n_c)). The term then is O(l*log(n_c) + l^2/n_c), which minimizes to O(l*log(l)).
> > >
> > > So the asymptotic complexity of the method we present is O(l*log(l)) rather than O(l*sqrt(l)). We believe that plane cutting hashes will yield the same experimental performance, but we will add the option to our implementation and verify that.

---

> > > > ### Public Comment · ~Aurko_Roy1 · 2019-10-02
> > > > **Thanks**
> > > >
> > > > Thanks for the update, makes sense!

---

### Public Comment · ~Hyunjik_Kim1 · 2019-10-02
**Choice of F and G for reversible attention?**

Hi, I have a question about the choice of F and G for reversible attention in Equations (7-9). So you choose F=Attention and G=FeedForward. Is this just to match the number of parameters with the original Transformer? Have you also tried F=Attention, G=Attention? If so, how does it compare? If not, do you expect this to be more expressive, perhaps at the cost of more parameters?

---

> ### Author Response · Authors · 2019-10-03
> **Choice of F and G for reversible layers**
>
> Thank you very much for your interest! Our choice of F and G is to maintain parity with the original Transformer, which allows us to verify that reversibility doesn't degrade model quality. There's a very large design space of alternative ratios between self-attention and feed-forward layers (as well as their relative order) that we didn't explore for this work.
>
> Regarding your question of F=Attention, G=Attention, it sounds like you're suggesting removing feed-forward layers from the model and replacing them with self-attention only. In our experience feed-forward layers are generally faster than attention, and making them wider is the most computationally-efficient way of increasing parameter count. LSH attention closes the asymptotic complexity gap between the two layer types, but feed-forward layers still have an edge in terms of constant factors.

---

### Public Comment · ~Benjamin_Börschinger1 · 2020-01-09
**Question about equation (2)**

Is it possible that equation (2) should read

  o_{i} = \sum_{j\in P_{i}} \exp{(q_{i} \cdot k_{j} - z(i, P_{i}))v_{j}}

i.e., that k_{i} should be k_{j}?  Same question for the repeated occurrences of equation (2) in the paper.

---

### Public Comment · ~Junhao_Wang3 · 2020-02-28
**Will pre-trained Reformer on large English corpus be released?**

Will pre-trained Reformer on large English corpus (like BERT) be released? And if so, what is the estimated timeline?

---

### Public Comment · ~James_Tian1 · 2020-05-26
**Sharing QK for sequence-to-sequence tasks**

Hi,

Very interesting paper! Just one question.

Given the transformer was originally a sequence-to-sequence model consisting of both an encoder and a decoder where the decoded message need not be the same length as the encoded message, how does the reformer still work in that circumstance? Wouldn't the fundamental difference between queries (which come from the decoder) and keys (which come from the encoder) there make sharing the QK space not possible? Put another way, doesn't sharing QK space limit you to self-attention?

Sometimes there is some ambiguity in terminology where e.g. BERT uses only the encoder blocks of a transformer and still calls itself a transformer. Is the reformer only reproducing the encoder blocks?

Thanks!

---

### Public Comment · ~Alexander_Mathiasen2 · 2022-06-06
**Does the Reformer have more parameters than the baseline?**

From paper:

page 2:
".. show that it performs the same as the normal Transformer when using the same number of parameters; we achieve this by having both x1 and x2 have size d_model. "

page 8:
" The two models have identical parameter counts, and the learning curves likewise appear to be nearly the same. "

I see how the parameters of Attention and MLP does not increase. But what about
(1) the embedding layer and
(2) the final projection layer?

Question 0. Why does the parameters of the initial embedding layer not increase if we double d_model?

Apologies for any misunderstanding.

---

> ### Author Response · Authors · 2022-06-06
> **Clarifying d_model**
>
> In reversible layers, we do not directly double d_model. Instead, at the beginning of the reversible block, we duplicate the d_model-sized vector, so x becomes [x, x] concatenated. This happens after the embedding and similarly we can reduce dimensionality before the final projection - so the parameter count is not affected.

---

> > ### Public Comment · ~Alexander_Mathiasen2 · 2022-06-07
> > **Final clarification**
> >
> > Thanks for clarifying. How do you reduce the dimension in the end? y2? sum(y1,y2)? Did you compare different approaches?

---

> > > ### Author Response · Authors · 2022-06-07
> > > **Clarification**
> > >
> > > I believe we tried both concatenation (which increses the final number of weights) and adding and we could not see the difference. I do not have access to these runs any more though - so it's just from my memory at this point, so please take it with a grain of salt.

---

### Decision · Program_Chairs · 2019-12-19

**Decision:**

Accept (Talk)

**Comment:**

Transformer models have proven to be quite successful when applied to a variety of ML tasks such as NLP.  However, the computational and memory requirements can at times be prohibitive, such as when dealing with long sequences.  This paper proposes locality-sensitive hashing to reduce the sequence-length complexity, as well as reversible residual layers to reduce storage requirements.  Experimental results confirm that the performance of Transformer models can be preserved even with these new efficiencies in place, and hence, this paper will likely have significant impact within the community.

Some relatively minor points notwithstanding, all reviewers voted for acceptance which is my recommendation as well.  Note that this paper was also vetted by several detailed external commenters.  In all cases the authors provided reasonable feedback, and the final revision of the work will surely be even stronger.